# The Health of Healthcare Professionals in Italian Oncology: An Analysis of Narrations through the M.A.D.I.T. Methodology

**DOI:** 10.3390/bs12050134

**Published:** 2022-05-05

**Authors:** Gian Piero Turchi, Ilaria Salvalaggio, Claudio Croce, Marta Silvia Dalla Riva, Luisa Orrù, Antonio Iudici

**Affiliations:** Department of Philosophy, Sociology, Pedagogy and Applied Psychology, University of Padua, 35131 Padua, Italy; ilaria.salvalaggio@studenti.unipd.it (I.S.); claudio.croce@studenti.unipd.it (C.C.); martasilvia.dallariva@unipd.it (M.S.D.R.); luisa.orru@unipd.it (L.O.); antonio.iudici@phd.unipd.it (A.I.)

**Keywords:** healthcare professional, oncology, health, burnout, psycho-oncology, interaction, discourse analysis, health promotion, dialogic science, cancer, M.A.D.I.T. methodology

## Abstract

From the analysis of the scientific literature relating to the health of oncological patients, the need to consider the global dimension of health of individuals emerges, which subsumes the bodily dimension and involves all the actors who offer their contribution to it in different ways. In this direction, the state of the art of the health construct offered by healthcare professionals highlights a lack of scientific contributions to the specific subject although these professionals are fundamental figures in oncological diagnosis setups. Considering, therefore, the healthcare roles as an integral part of the interactive framework where the oncological patient is placed, this paper offers the results of an Italian study relating to the health of healthcare professionals who take charge of patients with a neoplasia diagnosis. In particular, through an analysis of the discursive productions of 61 participants (healthcare workers, oncological patients and citizens) by the M.A.D.I.T. methodology (Methodology for the Analysis of Computerized Textual Data), this study aimed at observing the discursive reality of health offered by healthcare workers. The collected data highlight a low degree of health expressed by the healthcare professionals, who are strongly typified by rhetoric such as “the one who is destined to suffer psychologically”. These narrations limit the possibilities of development of different narrations in depicting these professionals: critical repercussions in the interaction with the oncological patients emerged, as well as in their global health degree. In conclusion, the results show the need for deep investigation into the impact that the health degree of health professionals can have on the patients they take charge of.

## 1. Introduction

Studies on the world effect of cancer diagnosis [1] affirm that one person out of five will develop cancer during his/her life, and that cancer is the first or second cause of death in 112 countries out of 183. The annual report of the American Cancer Society (ACS) and the International Agency for Research on Cancer (IARC) stated that in 2020, cancer deaths totaled around 10 million, while the number of new cases was calculated at about 19.3 million [1]. This is a datum growing fast throughout the world: it has been estimated that in 2040, there will be an increase in cancer cases by nearly 50%, thus reaching 28.4 million cases [1]. Such data offer a picture of the health status of the world population concerning the spread of neoplastic syndromes considering the bodily dimension, but what will be the impact of this disease on global patient wellbeing? 

In order to deepen the “health” construct, pivotal in this research, a philological deepening is useful. During the development of the current socio-cultural context, the usage value of the “health” construct has coincided with the “soundness” construct, and thus body integrity [2,3]. Despite that, as declared by the WHO (World Health Organization), health is a state of complete wellness that goes beyond the mere lack of disease/disorder [4]: it is a fundamental human right, whose achievement is the most important social aim at a global level, and “whose realization requires the action of many other economic and social sectors as well as the health one” [5]. In this regard, the scientific community continues to discuss the health notion: it provides a dynamic perspective of health that keeps in mind the social and relational aspects of citizens’ lives [6], who are in constant interaction with other members of the community to which they belong [7]. Thus, health is considered a construct that also involves the biological level, where the illness is considered within the person’s biography and managed through the support of the other roles involved [8]. Actually, a recent study in the oncological field [9] studied the competencies of patients and caregivers in the management of the implications of surgery: it emerged that oncological treatments have effects on all aspects of life; thus, the roles in an oncological situation need to develop new competencies in order to preserve life quality [10]. Moreover, discussing one’s own health implies interacting with people who listen, who in turn will talk about the health of other people (then, he/she will generate a specific narration about it) [11]. Considering the repercussions generated by a cancer diagnosis in the various areas of a patient’s life, the interactive framework will necessarily involve other roles (suffice to say how the patient will have to manage work issues, or family ones). Using the above-mentioned aspects within the present issue, Willing [12] described how the cancer experience is bound to the definition of it by others, and how this can either limit the biography or open other possibilities. Thereby, in order to speak about health, considering the interactive network that generates it is necessary, so that the various roles that participate in it can contribute to its generation and management [13,14]. 

Considering the cancer patient as part of an interactive framework is useful: the professional roles with which the patient and his/her caregivers interact to manage the health sphere are part of this framework, as well as family members and friends [15]. As they play roles that can be entrusted both with the health situation and the interactive repercussions, healthcare workers can intercept a critical situation and manage it, in order to promote the health of the patient [16]. Moreover, they configure themselves as playing a fundamental role in offering support to the caregivers of these patients [17,18], who are considered important parts in the management of the patient and of his/her new life, both at the physical and interactive levels, and, thus, in health [19,20]. 

Thus, if we consider health as a framework belonging to the interactive level, the health offered by healthcare professionals becomes a factor that influences the oncological situation of patients, and vice versa.

### 1.1. The Health Offered by Healthcare Professionals Who Work with Cancer Patients 

The literature shows little contribution concerning the health of healthcare workers in the oncological field. The analyzed studies focus on the burnout degree of these practitioners, linked to the relational dimension of the job environment [21,22,23,24]. Burnout syndrome is considered the consequence of a chronic stress related to work, which is the last stage of a defensive and reactive process to demanding work conditions at the emotional level [25]. In general, it is a defensive reaction incurred by those whose job is characterized by intense and frustrating interpersonal relations and professional responsibilities (in environments with ambivalent, conflictual and disappointing relations, and poor pay in particular [26]). Consistent with this definition, the Maslach Burnout Inventory (MBI) is the main tool to ascertain the presence of burnout, and it consists of three subscales: “emotional exhaustion”, “depersonalization” and “lack of personal accomplishment” (or “professional accomplishment”) [27]. Moreover, the World Health Organization has categorized burnout as “Problems related to life-management difficulty” and described it as a “state of vital exhaustion” in the International Classification of Diseases (ICD-10) [28]. Despite that, the lack of diagnostic criteria stresses the difficulty in defining the construct in a unique way. In order to make such a construct less vague, further details were presented at the 72nd Session of the World Health Organization’s (WHO) World Health Assembly held in 2019, which could be inserted in the ICD-11 (whose adoption by the member states will come into force from January 2022), in the chapter “Factors influencing health status or contact with health services”. Therefore, burnout could be considered a syndrome originating from chronic stress in the workplace which is not managed successfully. Within the manual, burnout is defined by three dimensions: feeling overwhelmed and without energy; increase in the mental distance from one’s own work, or feelings of negativism or cynicism about one’s own job; and reduced professional efficacy [29]. 

As already reported, within healthcare, the oncological sector is one of the most vulnerable to burnout syndrome development. Some international studies [30,31] have estimated that the presence of burnout among medical doctors and nurses in oncology wards runs at around 32%. A more recent study highlighted that 1 oncologist out of 10 shows significant symptoms. In a study by Allegra and colleagues [32], the presence of burnout signs in more than 60% of the sample, composed of 1740 oncologists of the U.S. medical community, was determined. With a multi-factor view, the authors found a direct proportion between the syndrome, the working hours needed for taking care of patients and administrative tasks.

Moreover, the most recent studies in health psychology stressed how the emergency pandemic situation, linked to the COVID-19 virus spread, has also had negative repercussions on the degree of burnout. Some studies [33,34,35] have shown that a greater workload (most of all, in environments with high risks of contagion) and the reduced availability of family members and colleagues to offer support in light of the health restrictions result in being significant in the burnout explanation, as well as the poor human and material resources (for instance, the initial lack of suitable PPE—personal protective equipments).

### 1.2. Aim of the Research

Consistent with what is evidenced in the literature, health involves a number of issues, including health and interactive/social issues: the reality of health, therefore, is generated and managed in the interaction between the roles involved (see Section 2). Consistently, the aim of such a study is to offer a description of the discursive modalities: according to the theoretical and methodological references adopted (see Section 2.1) in this study, the interactions are considered as dialogical interactions, and the interactive modalities implemented to create the health reality of the healthcare professional in the oncological field can be considered as discursive modalities. The latter are used not only by healthcare professionals, but also by their friends and relatives, oncological patients and other citizens, which generate narrations about health created and used by healthcare professionals in the oncological field. If the health of healthcare workers is observed with the degree of burnout presence, narrations related to such a construct will be necessarily available as a study object. Thus, speaking exclusively of burnout limits the possible narrations about the work of healthcare professionals who work with oncological patients: in this way, health is tied to such a construct, opening to rhetoric and narrations according to which being stressed does not allow considering oneself as healthy (therefore giving value to an overview of possibilities that the narrations about “stress” do not allow contemplating). The aim of this study was to describe the discursive modalities used by the involved roles that generate the narrated reality of the “health of the healthcare professional who works with oncological patients”. In such a way, we look at the current narrations on the subject, considering the references to burnout and the difficulties that arose because of the current pandemic, among others. This paper describes a study conducted in Italy, aimed at offering a description of the health narrations of healthcare workers who take charge of oncological patients, investigating how the professional role is described—the strengths and criticalities of this job, and how they can be managed—as well as the health promotion of workers and patients. This study is intended as an explorative work, to offer a contribution that remedies the lack of research in this specific area of study.

## 2. Materials and Methods

As mentioned above, within this study, the discursive modalities used by the speakers are important, since, in the theoretical–epistemological framework where this study is placed, the reality at issue is created through language. Starting from the contributions by Wittgenstein [36] and Salvini [37], Dialogic developed [3,13,38,39,40,41,42,43,44,45]: it is a scientific approach according to the sense of reality is created through interaction [46]. In fact, the theoretical background of this work is rooted in the interactionist perspective [13,47] and finds its place within the narrativistic paradigm in particular [3,11,48]. 

Consistent with these bases, this study used the M.A.D.I.T. methodology (Methodology for the Analysis of Computerized Textual Data) [13,14,15,16,38,43,44,45,46,47,48,49,50,51] to describe the health narrations of healthcare professionals in the oncological field. 

### 2.1. Measurements

In order to analyze the collected answers and describe the reality of the health of the healthcare professional who serves users with cancer (“Operatore che Presta Servizio ad Utenza con Neoplasia” in Italian; O.P.S.U.N. from now on), we used the M.A.D.I.T. methodology for text analyses. It considers content and process analysis: the first concerns the observation of the subjects (the content) that emerge from the answers of the participants of the study (methodologically defined as archipelagos of meaning (by “archipelagos of meaning”, we intend a type of independent content that contributes to repertory creation, and through which the repertory is organized in narrative terms (as a result of consistency) [43])); the second allows describing the discursive modalities (the ways) through which the content is offered in the text [13,14,15,16,43,44,45,46,47,48,49,50,51]. The discursive modalities are referred to as what Dialogic has called Discursive Repertories (D.R.) [2,3,9,11,13,14,15,16,43,44,45,46,47,48,49,51], defined as a “finite mode of creation of reality, linguistically understood, with pragmatic value, which also groups together several statements (called ‘archipelagos of meaning’), articulated in concatenated sentences and diffused with the value of assertion of truth, aimed at generating (configuring)/maintaining a narrative coherence” [43] (p. 13). Therefore, Discursive Repertories are elements that compose the reality under observation. Moreover, a priori, a measurement index is attributed to each D.R., which is the dialogic Weight (dW). The dW can have a value between 0.1 and 0.9 and is defined as the “capacity of the Repertory to create discursive configurations (i.e., realities) in relation to the group to which it belongs (generative, maintenance, hybrid)” [43] (p. 92): according to the specific process property, the Repertories are described in a glossary and are organized in the Semi-radial Periodic Table of Discursive Repertories (see related figure in [2,3,9,11,13,14,15,16,43,44,45,46,47,48,49,51]), which divides them into three types: generative, stabilization and hybrid repertories. The generative repertories category includes the use’s rules of the natural language that are characterized by the variability of the dialogic process and open to the possibility of generating a different sense of reality, other than the existing one. The stabilization repertories refer to discursive productions that create a “stable”, static sense of reality, which remains as it is, closed to change possibilities. The hybrid repertories category includes the use’s rules of natural language that increase the repertory capacity (to maintain or to generate) with which they are associated. A text with a high dialogic weight is characterized mainly by generative repertories that make the reality mutable and open to change. On the contrary, a text with a low dialogic weight is characterized mainly by stabilization repertories, which generate a stable and unchanged reality. Imagine, for instance, that the O.P.S.U.N. work is labelled as an exhausting job: if such rhetoric is channelled by stabilization repertories (low dialogic weight), the reality remains stable and the O.P.S.U.N. work remains exhausting as it is; if such rhetoric is channelled by generative repertories (high dialogic weight), the configuration is open to change and thereby the O.P.S.U.N. work could change and also become stimulating, valuable, etc. 

In the design of this study, we referred to the following health definition: “a configuration that is created within the dialogical process, that considers the occurrence of diseases and/or the generation of theories on the illness in anticipation” [3]. Anticipation is defined as a discursive modality that, starting from the current situation, generates multiple, different and uncertain situations that can happen, and that have not happened yet [43]. Hence, a high degree of health is observed when multiple possible scenarios are considered, regarding disease outbreaks (for instance, a neoplastic syndrome) and/or theories on the illness (for instance, rhetoric such as “if you have cancer your life ends”). Actually, this could allow for managing a number of biographical aspects that could be characterized as critical, such as neoplastic syndromes, before they happen. Anticipation then becomes a useful tool to limit the chance of a burnout diagnosis becoming predominant and unique in a person’s biography, instead of one of the many paths of life. Thus, the health degree varies as much as the generative value of the discursive production: the lower the dialogic weight of the sense of reality, the lower the health degree measured; vice versa, a high dialogic weight indicates a high health degree.

### 2.2. Participants

As previously stated, the object of investigation is the health narrated by healthcare professionals who work with oncological patients. Health is generated in the interaction between the discursive productions of the various roles involved in the oncological situation; thus, defining these roles is crucial. The sample of the study was therefore composed of doctors, nurses and social workers in public health service. 

In a preliminary phase, reflecting on the roles that can interact with the healthcare professionals, as well as other colleagues, a range of roles were identified, denominated as the “affective nucleus” of the worker. These include family members, partners or close friends of a professional who works with cancer patients and were included as part of the sample. This study also included cancer patients (recovered or not), both because they are the direct users of the service offered by healthcare professionals, and because health fits in an interactive framework which includes the professionals’ health as well. Lastly, involving citizens excluded from the previous categories was useful, in order to collect the discourses generated by the community in a broader sense. 

All participants explicitly agreed to the data processing for the purposes of this study. The sample consisted of 46 females and 15 males (*n* = 61), with an average age M = 40.51 (SD = 14.16), mainly with a single or married marital status (see Appendix A for further details of the sample). Half of the respondents obtained a bachelor’s degree or a master’s degree and, gradually with less frequency, a high school diploma, mandatory schooling, a professional qualification diploma and a postgraduate diploma. Considering the composition of the sample, 22 participants working as medical doctors, nurses and health and social care workers were included, who work or have worked with cancer patients (most of them have more than 20 years of experience or less than 5 years of experience); a total of 18 participants belonged to the categories “affective nucleus” and cancer patients (recovered or not), and 21 citizens were included.

### 2.3. Tools

In order to describe health in the way it is narrated by healthcare professionals in the oncological field, we designed a questionnaire to collect the discursive productions of the roles that create the reality of the workers’ health. To precisely observe the aspects characterizing the investigated reality, the questionnaire (Appendix A) was based on 10 questions, divided into 3 areas of content:How the narrative reality of the “healthcare professional who serves users with cancer” (“O.P.S.U.N.”) is described;Health promotion of the O.P.S.U.N.;Health promotion of cancer patients.

#### 2.3.1. Area A

With the first area, we collected the discursive productions that generate the O.P.S.U.N. reality, through questions that place the respondent in his/her professional role (question no. 6) or that make him/her imagine how another individual could narrate the healthcare professional role (question no.1). The questionnaire also included a question (no. 7) to observe the pervasiveness of the profession in other contexts of life. Through questions 2 and 4, we collected the critical aspects, including the difficulties that the worker faces during or after his/her shift, and the strengths, including the positive features of the O.P.S.U.N. job considered by the respondents. Question 3 discussed the merits of the critical aspects described in question no.2, collecting the modalities with which they are considered (for instance, if they are “insurmountable” rather than “manageable”).

#### 2.3.2. Area B

With the second area, we intended to explore health, describing how health promotion can be narrated not only by the O.P.S.U.N. role itself, but also by the other selected roles (see Section 2.2). As the key role of the study, collecting the narrations placing the respondents in the role played in relation to the O.P.S.U.N. job (affective nucleus, patients, citizens) and in the shoes of the same professional role was necessary. 

#### 2.3.3. Area C

With the third area, we collected the discursive productions about the health promotion of cancer patients. Therefore, it consisted of describing how the promotion of health is narrated, if it is possible or not and if it is linked to the sole responsibility of the O.P.S.U.N., of the patient, of both roles or of the whole community. Considering this area is useful, since the cancer patient’s health affects the health of the O.P.S.U.N., as already explained in the introduction.

## 3. Results

The description of the results obtained through the text analysis follows the various areas of investigation and, within each of them, what emerged from the three categories of roles involved (Table 1).

### 3.1. Area A: The Reality of the “Healthcare Professional Who Serves Users with Cancer”

Concerning the “O.P.S.U.N.” category, the discourses generated within the first area were characterized by a high frequency of stabilization’s discursive modalities, in particular by the Judgment Repertory (28.71%) and the Certify Reality Repertory (25.74%). As for the first one, discursive modalities that place a value, moral or quality aspect with the items introduced in the configuration were used. The content used is linked to the “suffering”, “psychological support” and “feelings” themes (“They were days of pain and infinite sadness, but also of great hope”, “The patients’ suffering here is greater than in other wards (…)”, “We are prepared to intervene on the disease, but way little on what is called emotional-psychological support, essential in these places”). As for the Certify Reality Repertory, discursive modalities that narrate the worker role in a stable and unchangeable way were used; examples are texts such as “need to be close [to the patients] but far, not to be overwhelmed”, and “work is always with you”. Then, the Description Repertory (19.80%) followed, which increased the value of the dialogic weight (0.4). This repertory conveys a discursive process that generates a sense of reality starting from an objective position, not bound to personal theories. As a result, the narrated reality is agreeable among more interlocutors, and items are added to the same reality that produce knowledge about the subject. For instance, “At each shift, patients’ distribution based on blood tests and therapies start, health status checking during the therapies and any remedies because of patient discomfort, and support by the doctor who follows the patient”. 

Even for the “affective nucleus and patients” category, Certify Reality (35.71%) and Judgment (22.62%) were the most used repertories, followed by Description (19.05%). Regarding the content used in the answers, within this respondent category, the difficulty to separate work from other life contexts was stressed, such as free time.

Lastly, as for the “other citizens” category, the most frequent repertories were the same as those above, but with a greater occurrence of the generative repertories unlike the previous respondents: Certify Reality, 26.67%; Description, 20.00%; Judgment, 15.83%.

In general, in all three cases, the most frequent discursive modality was one of those belonging to the stabilization type, either Certify Reality or Judgment. In fact, the dialogic weight was found to be below half of the continuum in a cross-cutting way (“O.P.S.U.N.” and “other citizens”: dW = 0.4; “affective nucleus and patients”: dW = 0.3). Therefore, an unchanging sense of reality takes shape, not contemplating the chance of alternative scenarios: the O.P.S.U.N. role is highly typified by the work-related tasks.

### 3.2. Area B: The O.P.S.U.N.’s Health Promotion

As for the “O.P.S.U.N.” category, the Certify Reality (43.75%), Description (16.67%) and Proposal (14.58%) Repertories characterized the narrations within the second area. The Proposal Repertory belongs to the generative type and is defined as a “discursive modality that creates narrated realities that are uncertain, possible and aimed at the management of what is required, according to modalities that refer to a goal”. Here are some examples: “A psychological support and a better distribution of the shifts would be useful, allowing the mental recovery”, and “To create a service aimed at the promotion of the oncological health, I would start with the criticalities considered as crucial by patients, workers and their supporting network, so to build a project that personally involves those who will be the users at a later time, as well”. As for the services that could be useful for the promotion of the O.P.S.U.N.’s health, the respondents considered psychological support, group meetings among colleagues and training for the management of particular criticalities as effective. 

Concerning the “affective nucleus and patients” category, Certify Reality (38.10%), Description (19.05%) and Judgment (14.29%) were the most used repertories. Additionally, according to these respondents, “psychological support” is mainly considered as a strategy to promote the worker’s “health”, followed by the chance to apply some changes at the organizational level (such as a reduction in work shifts) and offer training courses. 

Similar to the O.P.S.U.N. category, the “other citizens” category mostly used the Certify Reality (24.14%), Description (20.69%) and Proposal (13.79%) Repertories in this area as well.

For every role, the value of the dialogic weight was low (dW = 0.3–0.4); starting from here, the promotion of the O.P.S.U.N.’s health was narrated as stable and unchangeable. This was particularly enhanced for the O.P.S.U.N. role, where the dialogic weight of the sense of reality was dW = 0.3. On the contrary, also considering the presence of the hybrid repertories (linked to the generative ones), the narrations offered by “other citizens” generated a balanced reality between the generative and stabilization repertories (48.29% and 51.71%).

### 3.3. Area C: The Health Promotion of Cancer Patients

In the “O.P.S.U.N.” category, the Certify Reality and Description Repertories (30.43%), followed by the Proposal Repertory (13.04%), were found. In general, a substantial balance between the two types of discursive repertories was observed, with a slight tendency towards generative discursive modalities (dW = 0.5). According to the healthcare professionals, the health of patients who use (or that have used) their services can be actively increased by offering them psychological support, recreational activities also within the ward and attention to their feelings and to the way relations are made.

In this area, the category “affective nucleus and patients” used repertories belonging almost entirely to the stabilization typology, namely, Certify Reality (48%) and Judgment (28%). For this reason, the value of the dialogic weight was low (dW = 0.2).

As regards the “other citizens” respondents, the dialogic weight measured had a value of dW = 0.3; the stabilization typology was the most used, especially the Certify Reality Repertory (44.44%), followed by the generative Proposal and Description Repertories (14.81% and 11.11%, respectively).

According to the above-mentioned data, a clear difference between the realities of sense generated by the respondent categories was highlighted. The healthcare workers considered their role as an active part in the improvement of the patients’ health, but the second category did not give this same value. The “affective nucleus and patients” depicted the health degree as unchangeable. Thus, the category of the O.P.S.U.N. role mainly used discursive modalities that allow promoting health, and contemplating chances and strategies to modify and increase it. Vice versa, considering the low value of the dialogic weight (and thus the prevalence of stabilization repertories), we can affirm that the patients themselves did not consider their health as improvable, since it was narrated as unchanging and closely (and causally) linked to their disease. This aspect emerges from, for example, the following excerpt: “the uncertainty that surrounds the oncological disease and its treatment let the patient in a vortex of uncertainty and pain”.

In this way, rhetoric was perpetuated among the patients, by which health cannot be promoted on the basis of a certain health framework (such as the one related to the clinical picture of patients with neoplasia). Thus, the implication is that the health degree is not always improvable and possible, except under certain conditions: for instance, when there is also wholeness of the anatomical-functional unit (i.e., the body), i.e., when one is not ill.

## 4. Discussion

The gathered data and their analysis allowed us to obtain a description of the narrated reality of the “health of the healthcare professional who serves users with cancer” (O.P.S.U.N.), expressed by numerical values. Thus, in area A, concerning the reality of the healthcare professional, respondents expressed certainty and closure to change, limiting the life possibilities of the person exclusively to his/her job role. In other words, the healthcare professional role becomes extremely pervasive within the life of those who (also) carry out those duties. Actually, by observing how spare time was narrated by the O.P.S.U.N., it was described by all roles as bound to healthcare tasks: for instance, as “useful to put it aside for a bit”, to “vent what was accumulated” and/or as lived in a “better” or “worse” way because of one’s own job. Moreover, all respondent categories used the Cause Repertory, through which cause–effect linkages between carrying out a job and the psychological impact of interacting with treatable, but not curable, patients were highlighted. Thus, in the light of these results, those who offer their service to patients affected by cancer will be described as “the ones who are destined to psychological suffering”, keeping the reality even more stable and consistent with the prejudices of the historic-cultural context, as specified in the literature. Conversely, the use of generative narrations (as was done in some cases) makes “other” rhetoric available that are different from, for instance, “the need of psychological help”, “not knowing how to manage emotions” and “to be to an end”. Considering the construct of health as defined in Section 2, the sense of reality generated mainly by the use of generative repertories was characterized by a high value of the dialogic weight: this showed a high health degree, since it opens the reality to multiple biographical chances. Thus, once again, the anticipation of future scenarios allows placing an event (such as a burnout diagnosis) as one of the various aspects of life, without this being predominant and unique in the person’s biography. Carrying on with these scenarios, the worker could, for instance, use strategies such as “to cheer up the child I help, telling him a pun could put him a smile while he is here” and “having a beer with my colleague after work could be an occasion to ask him how he deals with certain difficulties”: the generated reality allows identifying suitable strategies to manage the critical aspects, in order to promote health. Instead, with the stabilization repertories, the various chances are not considered, but only one is regarded as possible. For instance, using repertories such as Cause and Delegating to Others, the possible result is processes that attribute responsibility to something outside, and thus there is no chance to anticipate different scenarios (one–one relation between cause and effect) or to manage what emerges because it is considered out of one’s own reach. Thus, citing the above-mentioned example, it is claimed that overcoming some criticalities is possible only with the help of an expert; thus, if an expert is not available, the situation will remain the same. Moreover, as for the promotion of the patients’ health, patients who suffer from a neoplastic syndrome perpetuate rhetoric according to which health cannot be promoted because of a given circumstance of health (such as that related to the syndromic clinical picture of patients with neoplasia). Thus, the implication is that health is not considered as a reality that can be promoted only under certain circumstances, for instance in this framework, when there is also wholeness of the anatomical-functional unit, i.e., when one is not ill.

On the other hand, the use of generative repertories, such as Description and Consideration, allows making room for discussions and sharing, and moving towards accountability in the management of the criticalities, by looking for new chances, in order to promote health. Above all, this class of repertories was actually used by the “other citizens” category” in area B and by the “O.P.S.U.N.” category in area C: as a result, the O.P.S.U.N. and citizens could be considered a resource, meaning that each role would feel directly and actively involved in the design of his/her own health, and the health of other people, as well as increasing its level using suitable strategies.

## 5. Conclusions

The analysis of the literature in the oncological field highlighted that few studies investigated the degree of burnout among healthcare professionals [52,53,54]. Thanks to this study, we have opened a new field of investigation: without a focus on a specific psychological construct, this study gives the opportunity to observe the reality of the health of healthcare professionals as narrated (and designed) by them and by the roles who talk about them [55]. In fact, considering what has been discussed thus far, burnout in the workplace does not necessarily imply a lower degree of health. It depends, indeed, on the way narrations about it are placed within one’s own life: If it becomes totalizing, dramatically limiting the modalities by which one describes himself/herself, then the health degree would significantly decrease, since strategies and chances of change have not been considered. On the contrary, if it becomes one of the multiple items composing one’s own life, it will have a lower impact since other roles and resources will remain open, and the person will be able to use them to create other possibilities, different from the burnout [45]. In particular, this study allowed observing that the health degree of the O.P.S.U.N. is generally quite low. The repertories (i.e., the narrative modalities) used by healthcare workers affect the way they interact with patients, family and friends: taking care of this professional category is necessary, since the low degree of health can generate fallouts in different areas, both in professional and personal life. Therefore, designing specific interventions is mandatory (for example, a specific training, or different forms of psychological support), in order to improve the health level of the O.P.S.U.N. using narrations that range from generative to hybrid and stabilization repertories. The use of all typologies of repertories allows the healthcare worker to open up the gaze on different ways of narrating his/her professional role and interact with other people, who can help him/her in being a resource for his/her job and, thus, for cancer patients.

As for the limitations of this study, the M.A.D.I.T. methodology for the text analysis requires technical-specific skills of the researcher, who has to be adequately trained. In order to increase the denomination process efficacy, machine learning techniques are being deepened and developed to carry out this practice [13]. Thanks to this technology, it will become possible to analyze a huge quantity of text data in a short time, keeping the margin of error under control, reducing the needed time and resources and extending the sample size. Another limitation, related to the sampling, is the non-distinction of the professional specializations of the participants (medical doctors, nurses, health and social care workers, etc): future research could also aim to independently observe how these figures contribute to the health degree of the O.P.S.U.N., and thus to assess and process anticipation and interventions even more precisely. Moreover, a sample consisting of women and men equally enables the formulation of more accurate descriptive statistics. Lastly, in this study, we highlighted how health involves the interactive framework where the citizen is located [6], keeping in mind the social and interactive aspects, and not only the ones linked to the bodily dimension. Therefore, because of that, it has been possible to maintain how the health of the cancer patient is linked to that of the professional who serves him/her [18], since they play roles that potentially have a decisive impact on the other’s life. This research did not go into detail on the impact that the professional’s health can have on the patient’s health, and vice versa: future research could aim at measuring the efficacy of the professional interventions of the healthcare roles in the narrations about the health of cancer patients. The data related to this dimension would allow designing intervention strategies ad hoc for the specific need identified in the interactive frameworks of the cancer field, which could have an effect on the whole community.

## Figures and Tables

**Table 1 behavsci-12-00134-t001:** Distribution of the repertory types by respondents’ category and by investigated area.

	“O.P.S.U.N.” Category	“Affective Nucleus and Patients” Category	“Other Citizens” Category
Area A: the narrated reality of O.P.S.U.N.	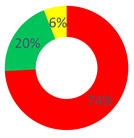	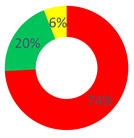	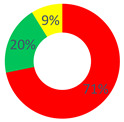
Area B: the health promotion of O.P.S.U.N.	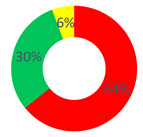	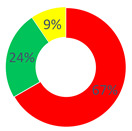	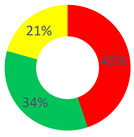
Area C: the health promotion of the cancer patient	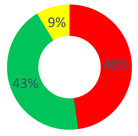	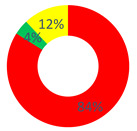	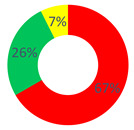

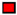
 Stabilization Repertories; 
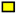
 Hybrid Repertories; 
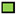
 Generative Repertories.

## Data Availability

The data presented in this study are available upon direct request to the corresponding author. The data are not publicly available due to privacy restrictions.

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
