# Peer review of "The Health of Healthcare Professionals in Italian Oncology: An Analysis of Narrations through the M.A.D.I.T. Methodology"

_behavsci, 2022, doi:10.3390/bs12050134_

Round 1

Reviewer 1 Report

This is an interesting topic and such a study could contribute valuable knowledge. However, I found this article extremely difficult to read due to extensive English language issues. This article needs significant English language editing before it can be usefully reviewed in English. There are many errors in sentence structure, word choice and grammar. There are so many that it is impossible to usefully highlight them all without essentially marking the article as though it is a student essay. Some phrases are not meaningful in English:

for example

Lines 43-44: “it is useful to do a philosophical lunging.”

Other sentences are very long and thus lack clear meaning: 

Lines 85-88: “From a careful analysis of the literature, it is brought to light the scarce scientific pro- 85 duction in terms of healthcare professional health, who takes care of the cancer patient 86 and if any, the studies focus on the burnout degree these workers undergo due to the relational environment where he/she carries out his/her duties”.

I therefore cannot provide a useful review of the content of this article in its current form. 

Author Response

We thank the reviewer for this observation: as suggested, an English revision of the whole paper has been carried out by an expert, in order to allow the review of the article. 

Reviewer 2 Report

Dear authors,

Thank you for providing the paper entitled "The Health of the Healthcare Professionals in Italian Oncology:  An Analysis of the Narrations through the M.A.D.I.T. Methodology". 

Psychological aspects of visiting cancer patients which finally can result in burnout are among the most area of investigations in this career that need special attention. And authors have provided a very well-conducted research and comprehensive review in this context. 

I believe the paper can be accepted as it stands. However, there are some minor revision to improve the quality of it. 

  1. providing a table depicting the demographic information of participants is essential. Data regarding the gender, current age, age at commencing their career in this field, work experience (oncology), daily business hours, night shifts, and etc. can be useful.
  2.  The conclusion is too lengthy. Please summarize it. 

Author Response

Thank for stressing the aspects mentioned above: as also the third reviewer asked, we shorten the conclusion paragraph, cutting some sentences that made it too long (see lines 509-529) and adding some aspects concerning the pragmatic use of the results. Also, we added a table of the demographic information of participants as supplementary materials, as suggested. Finally, an expert carried out an English review of the paper.

Reviewer 3 Report

This study deals with a very interesting and little studied topic. The originality of the work is noteworthy. 

I suggest that the discussion and conclusions section be rewritten to be more focused on the discussion regarding the results, but not to rewrite the results again, it makes for longer reading. 
In the study limits section, it would be convenient to refer to the fact that the sample is composed of more women than men, this variable may explain in some way the results, develop this limitation. 
In the conclusions, despite being an exploratory study, I miss more practical or applied implications that these results may have.

Author Response

As another reviewer highlighted criticalities about the length of conclusion, we edited both conclusion and discussion paragraphs, summarizing some aspects and cutting some sentences. Furthermore, we added the aspect concerning the composition of the sample as a study limit, and the possible practical implications of results in the everyday professional and personal life of O.P.SU.N.

Round 2

Reviewer 1 Report

Thank you for the opportunity to review this article again. I can see that significant editing has been undertaken and this has improved the clarity and coherence of the overall article. There are many sections highlighted in yellow - I am not sure what this means. Are these sections that have been edited? Further minor English editing would help with the flow of the writing even more. Some concepts are very complex so maximum clarity would be beneficial. There are also some inconsistencies in presentation of numbers (e.g. line 31  "one person out of 5" - write numbers in full for one to nine then use digits for 10 and over) and use of acronyms (e.g. OPSUN on lines 331 and 332). This all affects the flow of the writing.  

Author Response

Thanks for the revision, we appreciate the reviewer read the article again. We provided some linguistic corrections all over the paper, using the highlighted section as track. We modified the numbering as suggested, and clarified the OPSUN acronym in lines 163-164.